# The Association between Nursing Skill Mix and Patient Outcomes in a Mental Health Setting: An Observational Feasibility Study

**DOI:** 10.3390/ijerph20032715

**Published:** 2023-02-03

**Authors:** Nompilo Moyo, Martin Jones, Shaun Dennis, Karan Sharma, Michael McKeown, Richard Gray

**Affiliations:** 1School of Nursing and Midwifery, La Trobe University, Melbourne, VIC 3086, Australia; 2Victorian Tuberculosis Program, Melbourne Health, Melbourne, VIC 3000, Australia; 3Department of Rural Health, University of South Australia, Whyalla Norrie, SA 5608, Australia; 4IIMPACT in Health, University of South Australia, Adelaide, SA 5000, Australia; 5Whyalla Integrated Mental Health Service, Flinders & Upper North Local Health Network, Whyalla, SA 5600, Australia; 6Austin Health, Melbourne, VIC 3084, Australia

**Keywords:** readmission, mental health nurse, inpatient, nurse skill mix

## Abstract

Higher levels of educational preparation for nurses are associated with lower mortality rates in both medical and surgical wards. In mental health inpatient wards, few studies have examined whether specialist mental health nurse training has any impact on patient outcomes. The aim of this retrospective observational study was to establish the feasibility of extracting and linking nurse education and inpatient outcome data from hospital administrative sources to inform the design of future mental health nursing skill mix studies. Study participants were people experiencing mental ill-health and admitted to psychiatric inpatient care for at least 24 h. The exposure was the ratio of mental health nurses to comprehensive nurses for each patient for each day of their admission. The outcome was readmission for psychiatric inpatient care within 12 months of discharge from the index admission. Confounders were patient demographic (age, gender) and clinical characteristics (diagnosis, legal status, community follow-up). Forty-four patients included in the study were inpatients for a total of 595 days. The median hospital stay was 12 days (IQR = 7–17). In total, 11 (25%) patients were readmitted. In the readmitted and not readmitted groups, the median skill mix ratio was 5 (IQR = 5–7) and 5 (1–6), respectively. It was feasible to extract and code patient and nurse data from hospital databases and link them together. However, a substantial amount of manual post hoc recoding was required to enable us to calculate the exposure (mental health to comprehensive nurse ratio) in a precise way. It may be realistic to automate our methodology in an appropriately powered mental health nursing skill mix study. Australian and New Zealand clinical trial registry: ACTRN12619001337167p.

## 1. Introduction

Almost half of the mental health workforce are nurses [1], and they are the only professional group to provide care 24 h per day [2]. The training of nurses that work in inpatient mental health services differs in important ways between the health systems of different countries. For example, only registered mental health nurses that have completed specialist education in that field of practice can provide psychiatric care in the British National Health Service [3]. The training of nurses that work in mental health settings in Australia differs substantially from the UK system; nurses in Australia complete a comprehensive educational program at either the diploma level (enrolled nurses) or the undergraduate level (bachelor’s degree in nursing). Upon registration, they are authorised to work in any clinical setting, including mental health [4]. Although there are some differences between universities, comprehensive nurse training in Australia typically comprises one module of mental health nursing and three weeks of clinical placement [5,6]. Some registered nurses may choose to study for a specialist mental health nursing qualification, such as a graduate diploma in mental health nursing or a master of mental health nursing.

Readmission, HoNOS scores, absconding, forced medication, restraint and seclusion have been used as outcomes in mental health studies [7,8,9]. We choose readmission as an outcome measure because it severely impacts patients and their families while raising healthcare expenses [10]. In addition, readmission is common in psychiatric inpatient wards [11,12,13]. Zhou et al. [12] conducted a systematic review of the literature on unplanned hospital readmissions for patients with mental illness. The authors included one prospective study and 15 retrospective studies with a minimum sample size of 115 patients, and they reported that most studies were of moderate to strong quality [12]. Zhou et al. [12] reported that the 30-day unplanned hospital readmission rates ranged from 5% to 43%. 

The association between the educational preparation of nurses (one way of determining skills mix) and clinical outcomes for patients has been extensively studied in medical and surgical settings [14,15,16]. Skill mix can be defined as a proportion of staff experience and education [17]. Approaches to measuring skill mix include the proportion of registered nurses to unregistered nurses and the number of nursing hours per patient day [18,19,20]. Audet et al. [14] reported a systematic review of 27 observational studies (the total number of participants involved in these studies was not specified) that examined the association between nurse education and mortality in medical and surgical inpatient units. Because of the methodological heterogeneity of included studies, the authors did not undertake a meta-analysis. The authors of 16 of 27 included studies reported that a higher ratio of nurses with a nursing degree (baccalaureate) working in inpatient settings was associated with lower odds of patient mortality [14]. Although the authors report a critical appraisal of included studies—and considered studies to be generally rigorous—they did not use an established measure, and the statements about the methodological quality of included studies should be considered with caution. 

Skill mix research uses observational methodologies. Shin et al. [21] conducted a systematic review that included 19 studies, 12 of which employed a cross-sectional study design, and 7 were longitudinal. One of the limitations of cross-sectional studies is that specific nurses cannot be linked to individual patients, and researchers cannot be certain of the direct association between skill mix and outcomes [22]. 

Dall’Ora et al. [23] reported a systematic review of 27 observational studies on the association between nursing skill mix and patient outcomes in acute medical, surgical, and intensive care units. The authors report that almost all of the studies had a high risk of bias. A sensitivity analysis of studies with a low risk of bias indicated that higher staffing levels were associated with lower patient mortality [23].

In some studies conducted in medical, surgical and intensive care units, the authors extracted demographic and clinical patient data and nurse (roster and educational level) information from hospital databases and linked patients with the nurses who provided care [19,24,25,26,27]. Van den Heede et al. [26] undertook a cross-sectional study involving 9054 patients using two hospital data sources to assess the association between nurse staffing levels and in-hospital mortality in surgical wards. The authors reported that increasing nurse staffing was associated with a significant reduction in mortality and noted a significant association between the proportion of registered nurses with a Bachelor of Nursing degree and in-hospital mortality [26]. However, Van den Heede et al. [26] should have reported the number of nurses included in the study and how they linked the two data sources.

Yakusheva et al. [27] conducted a retrospective observational patient-level study including 8526 patients and 1477 nurses to examine the association between the proportion of care provided by a baccalaureate educated (BSN) nurses (BSN proportion) and patient mortality, readmission, length of stay, and inpatient expenses. The study was conducted in surgical and medical wards using data extracted from hospital electronic databases [27]. The authors reported that a 10% increase in the proportion of BSN-educated care was associated with an 11% decrease in the odds of mortality, whilst the associations with readmission, length of stay, and expenses were not statistically significant [27]. Manojlovich et al. [24] conducted a similar observational study to examine the association between nurse dose and methicillin-resistant Staphylococcus aureus (MRSA) infections and patient falls in 26 units using routinely collected hospital data. The authors reported that a greater proportion of registered nurses with a bachelor’s degree in nursing or higher were associated with decreased MRSA infections and patient falls [24].

Patricianet et al. [25] examined the association between nurse skill mix and adverse events at the shift level in medical-surgical and critical care units across 115,062 shifts from 13 military hospitals using patients and nurse data extracted from hospital databases. Patricianet et al. [25] reported that a higher proportion of registered nurses relative to unlicensed assistive personnel was significantly associated with decreased falls and medication administration errors. However, Patricianet et al. [25] did not consider the educational preparation of registered nurses; whether they had degrees in nursing. The findings from the studies conducted in the general, medical, and intensive care units indicate that the hospital administrative patient and nurse data can be used to examine the association between nursing skill mix and patient outcomes. However, these findings cannot be generalised to psychiatric inpatient wards.

The association between mental health nursing skill mix and patient outcomes has not been extensively studied. A systematic review of skill mix studies that have examined the association between mental health nurse to comprehensive nurse ratio and patient outcomes in mental health inpatient wards [28] reported no studies that met the inclusion criteria. Examining the nursing skill mix in mental health settings may be an important area for further enquiry. Prior to undertaking an appropriately powered observational study, it is first necessary to develop and test the feasibility of methodological approaches for skill mix research in a mental health setting. 

Feasibility studies are important because they may identify impediments to research and devise strategies to overcome them [29]. If adequately implemented, feasibility studies will facilitate the more efficient use of limited research resources for definitive studies and ensure that funds are granted to support research that can more likely be completed [29]. In addition, the publication of feasibility studies provides a convenient forum for discussing important parts of the scientific method, facilitating the exchange of knowledge in preparing for large scale clinical investigations [30]. 

### Objectives

The objective of this study was to examine the feasibility of extracting and linking the administrative patient and nurse data that would be required to test the association between mental health nurse skill mix and patient outcomes. 

## 2. Materials and Methods

We conducted an observational data linkage study. In this manuscript, we adhered to the Strengthening the Reporting of Observational Studies in Epidemiology (STROBE) guidelines (see Appendix A). As is best practice, the protocol for this feasibility study was pre-registered (before data collection started) on 30 September 2019 with the Australian and New Zealand clinical trial registry: ACTRN12619001337167p. The full protocol for this study has also been previously published [31].

### 2.1. Study Setting

The fieldwork for this study was conducted in two mental health services in Australia, one each in Victoria and South Australia. The South Australia site has one six-bed acute inpatient unit. The facility in Victoria has a 19-bed adult acute inpatient ward and 5-bed inpatient eating disorders unit.

### 2.2. Participants

#### 2.2.1. Patients 

In this study, we included all patients admitted to participating wards between 6 January 2020 and 5 March 2020, for at least 24 h, that were aged 18 years or older and were experiencing mental ill-health. 

#### 2.2.2. Nurses 

All registered nurses working on participating wards from 6 January 2020 through 5 March 2020 were included in the study.

### 2.3. Data Sources 

We extracted patient data for the South Australian hospital from the Integrated South Australian Activity Collection (ISAAC) and for the Victorian hospital from Cerner. In addition, nurse data for the South Australian hospital was extracted from the Complete Human Resource Information System [Chris21] and the paper shift roster, while data for the Victorian hospital was extracted from Kronos and the paper roster. Data sources for this study are described in more detail in the study protocol [24]. 

#### 2.3.1. Patient Data

We intended to extract the following demographic and clinical information: age (in years), gender, employment status, psychiatric diagnosis (ICD-10 codes), comorbidities, status under the mental health law, date of admission and discharge to the ward, Health of the National Outcome Scales (HoNOS) total scores on the day of admission and discharge to and from the ward. Additionally, we extracted the number and duration (in days) of admission(s) to the same inpatient service in the year following discharge from the index admission.

#### 2.3.2. Nurse Data

From the human resources databases, we extracted information about the number and educational qualification of nurses working in the participating wards for each day of the study. A day was a 24-h period starting at 07:00 (when the morning shift comes on duty) and concluding at 06:59 (when night staff go off duty); this was a pragmatic decision to reflect the ward routine in participating in clinical vices. Each registered nurse that participated in the study was coded as a mental health nurse if they had a formal mental health nursing qualification recorded in the human resources database. Additionally, we counted the number of enrolled (or Division 2) and agency (a nurse employed by a third-party organisation that may work across multiple different wards and hospitals) nurses who worked on the ward.

### 2.4. Ethics Considerations 

Approval for this study was received from the Victorian hospital’s Human Research Ethics Committee (ref: HREC/70480/Austin-2021). There were several important ethical issues that we considered when developing the methodology for this study. We applied for a waiver of consent on the grounds that—because we were using administrative data—the study was unlikely to cause distress to the participants [32]. Next, because data analysis was being undertaken at La Trobe University, it was important to ensure that no identifiable data were inadvertently shared outside of the participating clinical site. This required that we carefully consider the data that we requested and the procedures we used to deidentify. For example, we specifically did not extract data that included names, dates of birth, and addresses. As is common in studies of this nature, there are important ethical considerations in ensuring the security of the data. Extracted data were initially downloaded into an excel file and were checked by the site investigators at participating hospitals to determine that the correct information had been retrieved and that no identifiable information was included in the dataset. Once data were checked, it was uploaded into Cloudstor, which is a secure server that enables the secure transfer and storage of data [33].

The three participating organisations are part of the National Mutual Acceptance scheme. The National Mutual Acceptance scheme allows for the acceptance of a single scientific and ethical review of human research projects conducted in public health services across multiple centres [34]. After HREC approval, we applied for the site-specific assessment from both participating hospitals. The site-specific assessment is distinct from a formal ethical review in that it is focused on resources required for the project, the project budget, and site-specific policies and procedures [34]. Both hospitals approved the site-specific assessment applications (LNR 70480/2021 and 2021/SSA00366).

Finally, we applied for ethical approval from the Human Research Ethics Committees of La Trobe University, Ethics Approval—21/00632 to commence the study.

### 2.5. Statistical Methods

Excel files were imported into Stata/SE v15.0 (Stata Corp, College Station, TX, USA) for data checking and cleaning, following the procedures detailed in our protocol. We used the pairwise deletion method to manage missing data.

The exposure in this study was the mental health nurse to comprehensive nurse ratio, which was calculated from extracted data using the following procedure. We added all of the mental health nurses that worked in the ward during each patient’s hospital stay and divided that total with the sum of comprehensive nurses that worked in the ward during each patient’s admission. Table 1 is a worked example (not real patient data) of how we calculated this. An example of the linked patient and nurse data for patient X is available in Appendix A. 

To calculate the nursing hours per patient day, we divided a 24-h day by the number of patients per nurse [35]. 

Descriptive statistics were used to describe the demographic data and clinical characteristics of patients. We did not undertake an inferential analysis because the aim of the study was not to test the association between variables, but rather to establish the feasibility of extracting and linking patient and nurse data. 

### 2.6. Amendments to the Protocol

In our protocol, we proposed calculating the length of stay and nursing skill mix for each patient using averages; instead, we used the median because the hospital stay and skill mix distributions were skewed. Data that are skewed or not normally distributed should be reported as medians [36,37]. 

## 3. Results

### 3.1. Data Extraction and Linking

There were challenges in the data extraction process. The nurse roster data did not include qualifications. Extracting the nurse’s qualifications from the human resources databases and matching them with rostered nurses was complex and time consuming. There were no difficulties in extracting patient data. The fieldwork for this study took place during the COVID-19 pandemic, and there were delays in data extraction associated with the pandemic. Altogether, the patient and nurse data extraction process took us four months to complete. Linking the patient and nurse required calculating the nursing skill mix of each patient for each day of their admission, and it took us a median of 58 min (IQR = 16–60) to complete the process. The data were manually linked.

### 3.2. Data Checking

We conducted an initial check of the extracted data consistent with the procedures described in our protocol. We did not identify any apparent errors. HoNOS scores were not recorded for 4 (9%) and 8 (18%) patients on admission and discharge, respectively, because the ward clinical team had not completed the measure. We sought to extract information on the total number of individual nurses who worked in the study wards over the two-month period of the project. However, what was generated was the total number of nurses who had worked across the study period. Individual nurses were counted multiple times if they worked more than a single shift over the study period. By counting the number of individual nurses, we were able to generate a possibly more accurate measure of the ward staffing. 

HoNOS scores were missing for some patients. We conducted Little’s MCAR test to assess if data are missing completely at random (MCAR). Little’s MCAR test: Chi-Square = 1.560, Degrees of freedom = 2, *p*-value = 0.458. The test indicated that data are missing completely at random. We used the pairwise deletion method to manage the missing data as proposed in our protocol. 

The data for mental health nurse to comprehensive nurse ratio, hospital stay length, and HoNOS scores were not normally distributed. 

### 3.3. Participants

The flow of patients throughout the study is shown in Figure 1. Seventy-four patients were admitted to the participating wards, and 44 met our inclusion criteria. Thirty patients were excluded; of these, 27 were admitted before or discharged after the study period.

### 3.4. Demographic and Clinical Characteristics of Patients 

The demographic and clinical characteristics of patients are shown in Table 2. Most of the participants were males in the 18–35 age group. Nearly half had a diagnosis of either schizophrenia or mood and anxiety disorders. The majority (84%) of patients had at least one psychiatric comorbidity. The median hospital stays for all patients was 12 days (IQR = 7–17). Readmitted patients had higher HoNOS scores at discharge than those who were not. 

Patients in the readmitted group spent slightly fewer days on the ward than those who were not readmitted. 

### 3.5. Educational Preparation of Nurses

In total, 87 nurses were included in the study. Almost half (n = 41, 47%) had a formal qualification in mental health nursing, 24 (28%) were comprehensive nurses (with no mental health nurse qualification), 15 (17%) were enrolled (Division 2), and seven (8%) were agency nurses. 

### 3.6. Skill Mix (Exposure) 

At a patient level, 44 patients were exposed to a total of 12,723 nurses (mental health nurses = 7400 nurses, comprehensive nurses = 1896, enrolled nurses = 2166), and agency nurses = 1261) during the study period. The total number of patients who were in the wards across the study period was 1913 (total daily number of patients). Table 2 shows the median mental health nurse to comprehensive nurse ratio for all participants and readmission status. Patients who were readmitted during the study period had higher levels of input from mental health nurses compared to those who were not readmitted. The readmitted group had 2329 mental health nurses and 432 comprehensive nurses across the study period. The not readmitted groups had 5071 mental health nurses and 1464 comprehensive nurses across the study period. There was considerable variance in the nursing skill mix across included participants. 

The total number of nurses rostered across the study period was 1950; mental health nurses (n = 1135), comprehensive nurses (n = 312), enrolled nurses (n = 329), and agency nurses (n = 174). Appendix A shows the nurse-to-patient ratio during the study period. The nurse-to-patient ratio ranged from zero to one across the skill mix. The zero nurses to patients ratio does not imply that there were no nurses on the ward, but rather that one group of the skill mix had fewer nurses. There was a median of five comprehensive nurses and 32 patients (a median comprehensive nurse-to-patient ratio of zero when rounded off to the nearest whole number). The mental health nurse to patient ratio was one, and the nursing hours per patient day was 24 h.

### 3.7. Clinical Outcomes 

#### Readmission to Psychiatric Inpatient Care

Table 2 shows the number of patients readmitted to psychiatric inpatient care within one year of discharge. Of the 44 patients included in the study, 11 (25%) were readmitted. The median time from discharge to readmission was 101 days (IQR = 20–279). More than half of the patients that were readmitted had multiple readmissions (n = 8, 73%, range 2–4). Patients who were readmitted tended to be younger, male, have a diagnosis of schizophrenia and have been discharged on a community treatment order. 

## 4. Discussion 

The aim of this study was to test the feasibility of extracting and linking routine hospital administrative data that would be required to test the association between nursing skill mix and patient outcomes in an inpatient mental health setting. Whilst we were able to extract and link data, there were considerable practical challenges we faced when conducting the study, primarily related to calculating the nursing skill mix of each patient for each day of their admission. We had intended to automate this process, but because there were no common variables between the datasets, we had to manually code data, which was a complex, time-consuming process. We consider that it might be possible to automate the procedures we have described by including a common variable in both datasets in future studies. The nursing rosters in the participating wards did not indicate whether nurses had a mental health qualification, and it took a significant effort to add this information, which we recommend should be included in the rosters.

Ours is the first study to test the feasibility of measuring mental health nursing skill mix in psychiatric inpatient settings and linking it with patient outcomes. Authors of skill mix studies in general medical and surgical settings have shown that they are able to link nurse and patient data [15,16,27]. For example, Musy et al. [19] conducted a longitudinal study involving 128,484 patients and 4633 nurses to examine nursing care supply and demand, linked five datasheets at the patient, nurse, and unit levels in a single dataset using six key variables. In another observational study, Yakusheva et al. [27] extracted patient demographic, clinical data, multiple patient outcomes, and nursing skill mix data from hospital databases. In addition, the level of education of the nurses giving care on the wards was extracted from the hospital’s nursing administration database and linked with the patients who received their care [27].

In this feasibility study, we have achieved our goals. We plan to conduct a multicentre study across Australia to examine the association between the ratio of mental health nurses to comprehensive nurses and patient outcomes in psychiatric inpatient wards. We also recommend that similar studies be conducted in countries with comparable mental health nursing workforces to Australia.

## 5. Limitations 

Our study has important limitations that should be considered when appraising our work. Our methodology can only be applied in countries—such as Australia—where mental health inpatient wards are staffed by a mix of nurses with and without a specialist mental health nursing qualification. It would not be possible to apply this methodology, for example, in the UK, where inpatient units are only staffed by nurses with a mental health nursing qualification. Ultimately, this may restrict the generalisability of findings from a full-scale study using our methodology.

We were not able to extract data on several potential confounders that included nurses’ age and length of clinical experience; this was because the ethics committee that reviewed the protocol considered that these data were not necessary to answer the research question. A clear justification needs to be made in future studies as to why it is important to extract these data.

The other limitation is that patients were not allocated to a single nurse throughout their admission. For example, a patient receives care from a mental health nurse in the morning shift and is handed over to a comprehensive nurse in the afternoon. Our study did not consider the time nurses spend providing direct patient care, as this data is not routinely collected in the wards. In our protocol, we indicated that we would exclude inpatients that were under 18 years of age. However, some young people—under 18 years of age—are admitted to inpatient adult psychiatric units. It may be that the generalisability of our findings might be affected by excluding young people. In future studies, we suggest that no age limits should be applied.

HoNOS information was missing at admission or discharge for 10 (23%) patients in our study. The precision of effect estimations is reduced when missing data are present [38]. However, it is important to note that the HoNOS scales have been used in the past to measure the effectiveness of the mental health nursing care [7]. No patient had missing data on readmissions, indicating that this may be a reliable outcome. However, we did not determine how other outcomes, such as absconding, forced medication, restraint, and seclusion, were documented since we did not extract them, which is a limitation.

We did not collect data on the reasons for readmissions, and unclear if these data are collected. Future studies should consider collecting this information as a potential confounder. Some patients may be readmitted not because they relapsed, but for other reasons, e.g., to initiate clozapine treatment. In this study, only patients readmitted to the participating health services were included, resulting in an underestimate of the total number of readmitted patients. 

## 6. Conclusions

In this study, we have demonstrated that it is feasible to extract patient and nurse data from hospital databases and link them in a way that makes it possible to test the association between nurse skill mix and patient outcomes in mental health inpatient wards. Adequately powered studies that leverage our methodology are required to examine how nursing skill mix in mental health settings impacts patient outcomes. High-quality skill mix research is important to ensure that inpatient services are optimally staffed.

## Figures and Tables

**Figure 1 ijerph-20-02715-f001:**
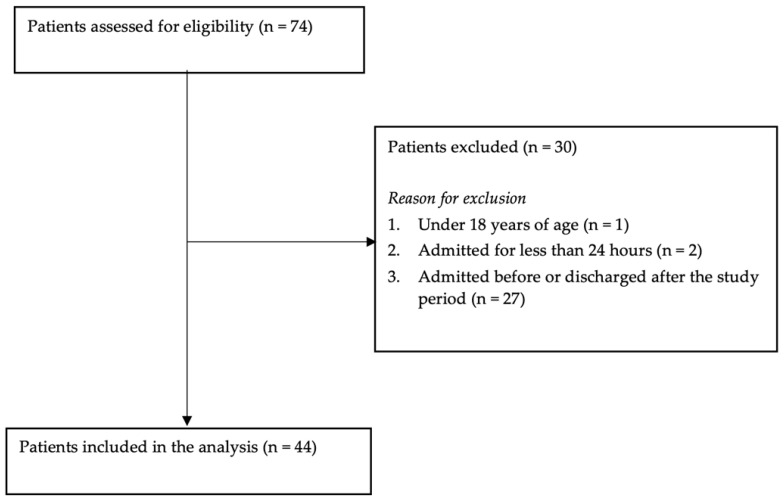
The flow of patients through the study.

**Table 1 ijerph-20-02715-t001:** The daily skill mix exposure for patient X for each day of their admission.

Skill Mix	1 20 January 2020	2 21 January 2020	3 22 January 2020	4 23 January 2020	5 24 January 2020	Totals	Mental Health Nurse to Comprehensive Nurse Ratio
Number of mental health nurses	3	4	5	4	4	20	2
Number of comprehensive nurses	2	1	2	3	2	10	

**Table 2 ijerph-20-02715-t002:** Demographic and clinical characteristics of patients.

Variable	All Participants N = 44	Readmitted within 12 Months N = 11	Not Readmitted within 12 Months N = 33
Age group in years (n, %)			
18–35	21 (48)	8 (73)	13 (39)
36–61	23 (52)	3 (27)	20 (61)
Gender (n, %)			
Male	28 (64)	5 (45)	23 (70)
Female	16 (36)	6 (55)	10 (30)
Other	0 (0)	0 (0)	0 (0)
Employment status (n, %)			
Employed	3 (7)	1 (9)	2 (6)
Unemployed	33 (75)	10 (91)	23 (70)
Unknown	8 (18)	0 (0)	8 (24)
Diagnosis (n, %)			
Mood/anxiety disorder	13 (30)	1 (9)	12 (36)
Schizophrenia	13 (30)	7 (64)	6 (18)
Other disorders	3 (7)	1 (9)	2 (6)
Any psychiatric comorbidity	37 (84)	11 (100)	26 (79)
Any physical comorbidity	30 (68)	10 (91)	20 (61)
Any substance uses	40 (91)	11 (100)	29 (88)
Admitted under the mental health law	29 (66)	7 (64)	22 (67)
Discharged on a community treatment order	10 (23)	5 (45)	5 (15)
Skill mix ratio * (Mental health nurse/comprehensive nurse; Median (IQR)	7400/1896; 5 (1–6)	2329/432; 5 (5–7)	5071/1464; 5 (1–6)
Hospital stays (in days) (Median (IQR)	12 (7–17)	10 (5–12)	14 (7–19)
HoNOS score at admission (Median (IQR)	15 (13–19)	15 (13–16)	15 (12–20)
HoNOS score at discharge (Median (IQR)	8 (5–11)	11 (9–12)	7 (5–10)

Notes. * Mental health nurse to comprehensive nurse ratio.

## Data Availability

The data are not publicly available because we did apply to do this during the ethics approval application and were unable to do it retrospectively.

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
