# Peer review of "The Association between Nursing Skill Mix and Patient Outcomes in a Mental Health Setting: An Observational Feasibility Study"

_ijerph, 2023, doi:10.3390/ijerph20032715_

Round 1
Reviewer 1 Report
This article is interesting in the sense that it is on the examine the feasibility of extracting and linking the administrative patient and nurse data that would be required to test the association between mental health nurse skill mix and patient outcomes. believe this paper will be important to the field, however, the novel finding of this article was not evident. In addtion, This manuscript looks like too much a summary of technical reports. I think the authors should wait until this work can be conducted in a more robust fashion.
Reviewer 2 Report
Please check the attached PDF file.

Reviewer 3 Report
Dear authors,
Your paper presents findings of a retrospective observational study that aimed to establish the feasibility of extracting and linking nurse education and inpatient outcome data from hospital administrative sources to inform the design of future mental health nursing skill mix studies.
The study found that it is still hazardous to extract and link patient and nurse data from hospital databases without a substantial amount of manual post-hoc recoding, which is required to calculate the exposure (mental health to comprehensive nurse ratio) precisely. It provides useful insights into the feasibility of using hospital administrative data to inform the design of future mental health nursing skill mix studies. The authors propose that it may be possible to automate their methodology in an appropriately powered study to overcome these obstacles. However, the literature review should be expanded to include more studies on the link between nursing training and clinical outcomes. Additionally, the paper's impact could be enhanced by providing better discussion on the implications for future research and recommendations for organizing mental health ward records and clinical/administrative databases.
With these revisions, the paper would be suitable for publication. It is recommended that the authors consider these suggestions before submitting the final version of the paper.
Given these limitations, I recommend is to study accept the paper for publication with minor revisions. The authors should consider the recommended changes before resubmitting the final version.
Round 2
Reviewer 1 Report
The authors have well revised their previous manuscript. I confirmed the author's opinion or idea of this manuscript. I recommend that it be accepted for publication.